# ESD Ideas: Translating historical extreme weather events into a warmer world

Ed Hawkins[1], Gilbert P. Compo[2,3] and Prashant D. Sardeshmukh[2,3]

[1] National Centre for Atmospheric Science, Department of Meteorology, University of Reading, Reading, UK
[2] Cooperative Institute for Research in Environmental Sciences, University of Colorado at Boulder, Boulder, USA
[3] NOAA Physical Sciences Laboratory, Boulder, USA

*Correspondence to*: Ed Hawkins (ed.hawkins@ncas.ac.uk)

**Abstract.** A new reanalysis-based approach is proposed to examine how reconstructions of extreme weather events differ in warmer or cooler counter-factual worlds. This approach offers a novel way to develop plausible storylines for some types of extreme event that other methods may not be suitable for. As proof-of-concept, a reanalysis of a severe windstorm that occurred in February 1903 is translated into a warmer world where it produces higher wind speeds and increased rainfall, suggesting that this storm would be more damaging if it occurred today rather than 120 years ago.

Whenever a severe weather event occurs with harmful impacts in a particular region, it is often asked by disaster responders, recovery planners, politicians and journalists whether climate change caused or affected the event. The harmful impacts are caused by the unusual weather, but climate change may have made the weather event more likely, more severe or both. In those cases, the harmful impacts may be partly or even mostly due to the change in climate. In some cases, the worst consequences may be due to the vulnerability or exposure of the local population or ecosystems, or a combination of many other factors (e.g. Otto et al., 2022).

Many methodologies exist to understand how climate change has affected extreme events. These are broadly categorized into risk-based approaches and storyline approaches (Stott and Christidis, 2023). The risk-based approaches assess the change in likelihood and magnitude of a particular class of event (e.g. Stott et al., 2004), whereas storyline approaches consider how climate change may have affected a specific event (Trenberth et al., 2015; Shepherd et al., 2018). Other studies consider the related question of what a plausible worst-case event might look like in a particular climate (e.g. Thompson et al., 2017). Some of these methods are now regularly used to provide attribution statements soon after events occur (van Oldenborgh et al., 2021).

Event storyline approaches attempt to quantify how an extreme event would be different in an altered climate. This can be achieved by producing reconstructions of the event as it occurred and in counter-factual cooler or warmer climates and comparing the consequences. Various approaches exist for such analyses, including statistical methods (e.g. Cattiaux et al., 2010), analogues (Ginesta et al., 2023; Faranda et al., 2022), nudging a weather or climate model (e.g. Meredith et al., 2015; van Garderen et al., 2021; Sánchez-Benítez et al., 2022) and forecast-based (e.g. Wehner et al., 2019; Leach et al., 2021). Here, we propose a complementary reanalysis-based approach to translate extreme events into different climates.

Specifically, we use the NOAA-CIRES-DOE 20th Century Reanalysis v3 system (20CRv3; Slivinski et al., 2021) which assimilates surface pressure observations into a NOAA weather forecast model to reconstruct the atmospheric circulation from 1806-2015 at 0.7° resolution using 80 ensemble members. Boundary conditions of SSTs and sea ice concentration are prescribed in 20CRv3. We generate counter-factual reconstructions of extreme events by rerunning the reanalysis for those events with perturbed SST boundary conditions and assimilating the same surface pressure observations. As atmospheric temperature changes will be primarily mediated by imposed changes to SSTs (rather than by direct radiative forcing; Compo and Sardeshmukh, 2009), this perturbation translates the reanalysis of the events into a changed climate.

As proof-of-concept, we have applied this approach to a severe extra-tropical windstorm known as Storm Ulysses, which occurred in February 1903, to examine how the consequences of this historical event may have been different had it occurred in a warmer world. Three experiments with the same reanalysis system are considered: (1) the original 20CRv3, (2) an improved version of 20CRv3 with added historical surface pressure observations and a small change to the data assimilation scheme (Hawkins et al. 2022), and (3) same as (2), but with a spatially uniform +2K perturbation added to the sea surface temperature (SST) boundary conditions. Sea ice, atmospheric greenhouse gas and aerosol concentrations are unchanged. A 6 month 'spin-up' period was run prior to the windstorm to ensure that the atmosphere had adjusted to the counter-factual warmer SSTs. Experiments are planned to explore the sensitivity of the findings to all these different choices. For example, we plan to repeat the simulations with atmospheric greenhouse gas composition representative of a +2K warmer world.

Storm Ulysses is not well represented in the original 20CRv3, but the improved reanalysis shows a far more intense storm, mainly from adding new rescued pressure observations; it is a credible reconstruction of one of the most extreme windstorms to occur over the British & Irish Isles in the past 120 years (Hawkins et al., 2022). Figure 1 (top row) shows the wet-bulb potential temperature at 850 hPa for one specific time during the storm for the different experiments. The storm in the improved reanalysis has a hook-shaped Shapiro-Keyser-type structure which is not present in the original 20CRv3. As expected, the structure of the storm in the warmer world reanalysis is constrained to be very similar to the improved reanalysis, except for a roughly 2.5K offset. Specific humidity and surface air temperatures are also higher (not shown).

The warmer world reanalysis of the storm generates stronger maximum winds (middle row) and increased rainfall (bottom row) compared to the improved reanalysis, suggesting that the consequences of the same surface circulation pattern would be more severe in a warmer world. The reconstructed circulation has some limited flexibility to vary given the observational coverage and uncertainties. While the increase in wind footprint appears small over land (around 1 m/s in some places), it becomes a significant increase in damage (>10%; roughly related to the cube of the wind speed; Klawa and Ulbrich, 2003). The total rainfall over land during the storm increases by 26%, i.e. around 10%/K when using the temperature change at 850

hPa to normalise the rainfall change, which is larger than might be expected due to the Clausius-Clapeyron relationship alone. An animation of the storm is included as a Supplementary Figure.

These experiments have reconstructed an intense windstorm in different climates and demonstrated the potential of using a surface pressure based reanalysis to develop heavily-constrained storylines which examine the thermodynamic and local dynamical changes in extreme weather events. A wider range of events and experimental designs need to be considered to examine the robustness and implications of the findings presented, and to build understanding about the types of weather event for which this approach is best suited. Downscaling these simulations would also be possible to provide additional smaller-scale details. Although this approach does not currently consider any large-scale dynamical component to the changes, it may be of interest to policymakers as extreme historical (and modern) weather events could be translated into warmer climates to construct examples of plausible 'worst-case' events to aid planning. This approach also offers the opportunity to attribute changes in observed weather events to human influence by translating reanalyses of modern events into cooler worlds. The 20CRv3 reanalysis currently ends in 2015 but could be extended to run in near-real-time to examine extreme events soon after they occur.

## Acknowledgements

Valuable discussions with Philip Brohan, Andrew Schurer, Ted Shepherd, Laura Slivinski, and Rowan Sutton are gratefully acknowledged. We thank Davide Faranda, Joseph Barsugli and an anonymous reviewer for their suggestions which helped improve the paper. Support for the Twentieth Century Reanalysis Project version 3 dataset is provided by the U.S. Department of Energy, Office of Science Biological and Environmental Research (BER), by the National Oceanic and Atmospheric Administration Climate Program Office, and by the NOAA Physical Sciences Laboratory. This research used resources of the National Energy Research Scientific Computing Center (NERSC), a U.S. Department of Energy Office of Science User Facility located at Lawrence Berkeley National Laboratory, operated under Contract No. DE-AC02-05CH11231. EH is supported by the NERC GloSAT project (NE/S015574/1) and by the National Centre for Atmospheric Science. GPC and PDS were supported in part by the NOAA Cooperative Agreement with CIRES, NA22OAR4320151, and by NOAA's Physical Sciences Laboratory.

## Data availability

Data available from: https://github.com/ed-hawkins/ulysses-storm-data

## Author contribution

Ed Hawkins: Investigation, Formal Analysis, Writing – Original Draft, Conceptualization

Gilbert Compo: Conceptualization, Writing – Review & Editing, Resources

Prashant Sardeshmukh: Conceptualization, Writing – Review & Editing

## Competing interests

None

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

# Storm Ulysses

26-27 February 1903

**Figure 1: Translating Storm Ulysses into a warmer world.** The columns represent the original 20th Century Reanalysis v3 (20CRv3; left), the improved reanalysis (middle) and the warmer world reanalysis (right). The wet-bulb potential temperature ($\theta_w$) at 850 hPa at 06 UTC on 27th February 1903 (top row) shows the synoptic structure of the storm as it passes over the UK & Ireland. 20CRv3 does not accurately represent the intensity of the storm, but this is corrected in the improved reanalysis, mainly due to adding additional observations (Hawkins et al. 2022). The warmer world reanalysis shows a very similar atmospheric structure to the improved reanalysis and is warmer by about 2.5K at 850 hPa. The thick lines indicate particular isotherms as labelled and the black dot represents the centre of the storm. The wind footprint (middle row; maximum ensemble mean wind speed at 10m during the storm) shows stronger peak wind speeds in the improved reanalysis than 20CRv3, which are stronger again in a warmer world. The track of Storm Ulysses is shown with the thick black line. The rainfall during the two days of the storm increases in a warmer world (bottom row). Supplementary Figure 1 is an animated version of this figure which shows the 850 hPa wet-bulb potential temperature, wind speed (with sea level pressure) and rainfall rate, every 3-hours during the storm for the three experiments.