# Peer review of "ESD Ideas: Translating historical extreme weather events into a warmer world"

_EGUsphere, 2023_

## Author Response (AR1)

**Translating historical extreme weather events into a warmer world**

**Preprint discussion:** https://egusphere.copernicus.org/preprints/2023/egusphere-2023-665/

To the Editor: the manuscript is slightly longer than the article type limits but we hope that this is acceptable given the need to fully address the reviewer comments.

To the Reviewers: we thank the reviewers for their helpful comments which we discuss below. We have made several other small changes to the text to improve clarity. In addition, we have corrected the figure which had included panels where the rainfall had been summed over the incorrect time period. This has now been fixed and the corrected version is shown here and included in the paper. The values given in the text were correct.

[Figure]

26-27 February 1903

**Joseph Barsugli**

I found this paper very intriguing and potentially a valuable addition to attribution studies. I have a couple of comments regarding the statement:

*"Boundary conditions of SSTs and sea ice concentration are prescribed in 20CRv3, and so counter-factual versions of extreme events can be generated by perturbing the SST boundary conditions and assimilating the same pressure observations. By using 20CRv3 no other perturbations are required, unlike if this approach used a reanalysis that assimilated other types of observation"*

Is the reasoning that no other perturbations are required due to the fact that a) atmospheric mass does not change (much) with time or with global warming, and b) a sizeable amount of climate change is mediated by oceanic temperature and concomitant humidity changes rather than by direct radiative forcing (as two of the authors have previously demonstrated). However I think that changing the atmospheric composition could also lead to additional radiative forcing over land over the 6 month spin-up time that would have an effect, albeit rather small on this storm. Perhaps more significant would be aerosol changes, not only for their radiative effect, but also for their effect on precipitation.

We thank the reviewer for their comments. Our reasoning for our experimental design is as the comment suggests: that the oceanic temperature perturbations will drive increases in atmospheric temperature and humidity over the spin-up period. We now cite the Compo & Sardeshmukh (2009) paper and discuss this briefly.

We also agree that there may also be indirect effects of atmospheric composition for some types of events, and we plan to explore these in further work. It is not possible to adjust non-volcanic aerosol concentrations in the current version of the 20CRv3 system as they are fixed at a present-day climatology for all years (Slivinski et al. 2019).

I also have a question about how surface winds over areas of dense observations would change using this methodology. Surface wind speed is relatively strongly constrained by pressure gradients and boundary layer stability. It would seem that constraining the surface pressure field with 1903 pressure observations would yield similar surface winds no matter the changes aloft or in SSTs at least in areas of denser observations. This would be consistent with the relatively small changes in wind speeds over land seen in the results. Perhaps there is something about this method, or about some unstated assumptions that I do not understand.

Although the atmospheric circulation is well constrained by the available observations, it is not so tightly constrained as to be identical in the two experiments. Between the uncertainty in the observations and the time-varying, flow-dependent uncertainty in the first guess, there is flexibility in the assimilation to allow some differences in pressure gradients, and hence surface winds, to appear. We plan additional experiments with other windstorms in the modern era where the circulation is more tightly constrained to see if this particular finding is robust across many events.

**Davide Faranda (reviewer)**

The paper "ESD Ideas: Translating historical extreme weather events into a warmer world" proposes a new approach to examining how extreme weather events may differ in a warmer or cooler world using a reanalysis-based approach. This approach involves producing

reconstructions of events as they occurred and in a counter-factual warmer or cooler climate to compare the consequences. The authors use the NOAA-CIRES-DOE 20th Century Reanalysis v3 system, which assimilates surface pressure observations into a NOAA weather forecast model to reconstruct the atmospheric circulation from 1806-2015. As a proof-of-concept, the authors translated the reanalysis of a severe windstorm that occurred in February 1903 into a warmer world where it produces higher wind speeds and increased rainfall.

The paper suggests that this approach offers a novel way to develop plausible storylines for some types of extreme events that other methods may not be suitable for. The paper also discusses the different methodologies that exist to understand how climate change has affected extreme events, such as risk-based approaches and storyline approaches, and how event storyline approaches attempt to quantify how an extreme event would be different in an altered climate.

Overall, the paper presents an innovative and interesting approach to examining extreme weather events in a changing climate. While I really appreciate the purpose of the proof of concept, the following points should be discussed in details in the revisions:

We thank the reviewer for their helpful comments and agree with all the suggestions that more details could be added. We have added some additional text and references to address the reviewer's comments, but we are constrained by the journal article type to around 1000 words and 15 references (which we slightly exceed).

-Forgotten Similar Approaches: there has been a recent attention on storyline approaches focused on analogues that look quite similar to those proposed here, as they make use of sea-level pressure patterns and seek for spatial changes of extratropical storms intensity: "A framework for attributing explosive cyclones to climate change: the case study of Alex storm 2020" by Ginesta et al. (2022, Climate Dynamics): This paper focuses on attributing the storm Alex, an explosive extratropical cyclone that hit Southern France and Northern Italy in October 2020, to climate change. The authors use the analogues method on sea-level pressure maps to identify 30 cyclones that match the dynamical structure of Alex for two periods: 1950-1985 and 1985-2021. They find that in the factual period, the anticyclonic circulation over the North Atlantic and the cyclonic circulation over Northern Africa are deeper than in the counterfactual period, and storms like Alex are more persistent and more predictable in present-like conditions. The authors also track the analogues and find that under current conditions patterns like Alex are more wavy than in the past. "Climate-change attribution retrospective of some impactful weather extremes of 2021" by Faranda et al. (2022 WCD): This paper addresses the question of whether climate change may have affected the characteristics of specific extreme events that occurred in 2021 over Europe and North America. The authors use the ERA5 dataset from 1950 to the present to define present (factual world) and past climate conditions (counterfactual world), and they identify the most similar sea-level pressure patterns to the extreme events of interest in each period. Other approaches that are similar but do not require to run simulations are those via stochastic weather generators introduced by Yiou, P.: AnaWEGE: a weather generator based on analogues of atmospheric circulation, Geosci. Model Dev., 7, 531-543, 2014. I think that these approaches should be mentioned at least as an alternatives of running every time ad-hoc simulations of each events.

We thank the reviewer for reminding us of these papers. We have added citations to two of these papers and highlighted the analogue approach specifically. We have also removed a comment suggesting that windstorms had not been looked at previously in event storylines.

-Limited scope: The paper only presents a proof-of-concept analysis of a single extreme weather event, and the approach has not been applied to a wide range of events. Therefore, the generalizability of the approach is still unknown, and further research is needed to validate its usefulness across a broader range of events.

We fully agree – this is a proof-of-concept to promote discussion about the types of event that may be suitable for analysis using this approach. This fits the aims of this article type within the journal. A sentence has been added in the final paragraph to highlight this.

-Assumptions: The approach relies on a number of assumptions, such as the uniformity of the sea surface temperature perturbation and the absence of other perturbations given by internal variability of the climate system. These assumptions may not hold true in all cases and could potentially affect the results. For example what would happen if the major source of moisture for the storm relies on an atmospheric river whose origin is somewhere by the tropics ?

We fully agree – in this proof-of-concept we chose a uniform SST perturbation to avoid changing the SST gradients along the storm's path, but it may be more appropriate to consider a spatially varying perturbation for other events. Further experiments are planned to explore this issue and the text has been edited to emphasise this.

-Interpretation: The paper does not provide a detailed discussion of how the results should be interpreted in terms of their policy or societal implications. It is important to consider how these findings could be applied to mitigate the impacts of extreme weather events in the future.

We fully agree. The aim of this study is to describe the conceptual framework to explore these issues in more detail. The initial idea is that this approach could be used to develop plausible high-impact events in the current or near-future climate to inform decisions. It would also potentially allow attribution of recent extreme events and their impacts by translating them to cooler climates. The final paragraph has been edited to highlight this issue.

-Uncertainty: While the approach provides a new way to understand the potential impacts of extreme weather events in a warmer world, it does not account for all sources of uncertainty. Therefore, it is important to acknowledge that the results are not deterministic and that there is still significant uncertainty associated with projecting the impacts of climate change on extreme weather events.

We fully agree. We plan to explore other events and perform sensitivity studies to quantify some of the uncertainties.

My reviews are always intentionally signed because I value an honest and transparent discussion between authors and reviewers.

We appreciate the honest and transparent discussion.

**Reviewer 2**

This manuscript investigates how Storm Ulysses (in 1903) would look like if it occurred in present-day climate conditions. The study is based on the latest version of the 20CR reanalysis.

The approach followed by the authors is interesting and promising, but I have some reservations on a few points.

1. The authors present the use of 20CR as a methodology or an approach. In my opinion, using this reanalysis is a way (among others) to construct analogs of a past storm. A "methodology" is the sequence of operations that uses reanalyses (or any other dataset) to produce a result, not the reanalysis itself, which exists independently of extreme event attribution issues.

We strongly disagree with the reviewer that our reanalysis approach is a way of constructing analogs. The reanalysis produces a reconstruction of the storm constrained by the available observations. The cited Hawkins et al. (2022) paper describes the improved reanalysis of Storm Ulysses and demonstrates its credibility in representing its features by comparing with independent data. Rerunning the reanalysis system with changed boundary conditions (warmer SSTs) to translate this storm into a warmer world is a new approach to understanding changes in extreme weather.

2. The idea of simulating analogs of observed storms in different climate conditions has already been published (e.g., among others, Ginesta, M., et al., A methodology for attributing severe extratropical cyclones to climate change based on reanalysis data: the case study of storm Alex 2020. *Clim Dyn* (2022). https://doi.org/10.1007/s00382-022-06565-x), albeit with another reanalysis.

We have added a citation to the Ginesta et al. paper, but that is an entirely different (statistical) approach which constructs a linear combination of similar sea level pressure patterns (analogs) to roughly mimic the storm in question and examine the different consequences in two periods. It does not involve rerunning the reanalysis used (ERA5) as we have done with 20CRv3.

3. The authors spend many lines reviewing (albeit partially) extreme event attribution, but barely explain how they obtain their figures.

We have moved some text around in the revised version but use just 13 lines to provide a brief review of the previous literature, 15 lines to describe the reanalysis system used and the experimental design, and another 15 lines to describe the figure and the change in consequences due to the storm being reconstructed in a warmer world.

Therefore, my feeling is that the authors present a new (and certainly great) dataset that is useful for extreme event attribution. But I don't see any conceptual advance for the methodologies of extreme event attribution. The authors could consider revising the manuscript by a more thorough bibliographic search, and a clarification of the methodology they actually use.

We disagree with the reviewer – this is a novel approach to understanding how reanalyses of extreme events change in a warmer or cooler climate. We have added additional references, but we are constrained by the article type limits. We feel this is sufficient given the length constraints for this type of journal article.

---

## Author Response (AR2)

**To the Editor**

In response to the final reviewer suggestions we have:
1)  added one sentence to highlight more explicitly that we plan to repeat the simulations with changed atmospheric composition, and
2)  reordered one sentence to make it clearer that we have rerun the reanalysis with changed boundary conditions for the same event.

We have also:
3)  added a neglected reference to Meredith et al. (2015) which used a nudging approach to look at an extreme event earlier than the example previously given, and
4)  added a thanks to the reviewers & a grant code into the acknowledgements.